# The Role of Cilostazol, a Phosphodiesterase-3 Inhibitor, in the Development of Atherosclerosis and Vascular Biology: A Review with Meta-Analysis

**DOI:** 10.3390/ijms25052593

**Published:** 2024-02-23

**Authors:** Minji Sohn, Soo Lim

**Affiliations:** Department of Internal Medicine, Seoul National University Bundang Hospital, Seoul National University College of Medicine, Seongnam 13620, Republic of Korea; rainbowmjs@naver.com

**Keywords:** atherosclerosis, cardiovascular diseases, cilostazol, platelet aggregation inhibitors, aspirin

## Abstract

Atherosclerotic cardiovascular disease (ASCVD) stands as the leading global cause of mortality. Addressing this vital and pervasive condition requires a multifaceted approach, in which antiplatelet intervention plays a pivotal role, together with antihypertensive, antidiabetic, and lipid-lowering therapies. Among the antiplatelet agents available currently, cilostazol, a phosphodiesterase-3 inhibitor, offers a spectrum of pharmacological effects. These encompass vasodilation, the impediment of platelet activation and aggregation, thrombosis inhibition, limb blood flow augmentation, lipid profile enhancement through triglyceride reduction and high-density lipoprotein cholesterol elevation, and the suppression of vascular smooth muscle cell proliferation. However, the role of cilostazol has not been clearly documented in many guidelines for ASCVD. We comprehensively reviewed the cardiovascular effects of cilostazol within randomized clinical trials that compared it to control or active agents and involved individuals with previous coronary artery disease or stroke, as well as those with no previous history of such conditions. Our approach demonstrated that the administration of cilostazol effectively reduced adverse cardiovascular events, although there was less evidence regarding its impact on myocardial infarction. Most studies have consistently reported its favorable effects in reducing intermittent claudication and enhancing ambulatory capacity in patients with peripheral arterial disease. Furthermore, cilostazol has shown promise in mitigating restenosis following coronary stent implantation in patients with acute coronary syndrome. While research from more diverse regions is still needed, our findings shed light on the broader implications of cilostazol in the context of atherosclerosis and vascular biology, particularly for individuals at high risk of ASCVD.

## 1. Introduction

Cardiovascular diseases (CVDs) [1], including coronary artery disease (CAD), cerebrovascular disease, and peripheral artery disease (PAD), are a significant global public health problem [1,2]. Ischemic heart disease is the most common cause of death worldwide [3]. Moreover, cerebrovascular diseases are more prevalent among Asian ethnicities, thus becoming a top priority for intervention. Since 2015, stroke has become the leading cause of death and disability in China, posing a significant public health threat [4]. Atherosclerosis in peripheral arteries is characterized by stenosis or the occlusion of the arteries that supply blood to the lower limbs, resulting in symptoms such as intermittent claudication and ulceration [2]. PAD is also related to a high risk of atherosclerotic cardiovascular disease ASCVD, including myocardial infarction, stroke, and renovascular disease [5]. About 1–3% patients with intermittent claudication exhibit progression to severe limb-related complications within 5 years, which may necessitate limb amputation and result in premature death [5].

These vascular diseases are mainly caused by atherosclerosis, which starts with the development of fibrofatty lesions known as atherosclerotic plaques. Several mechanisms are involved in this process: the accumulation of modified low-density lipoprotein (LDL) in the intima, the attraction of pro-inflammatory monocytes and T lymphocytes to the arterial wall, the proliferation and migration of vascular smooth muscle cells (VSMCs), the aggravation of oxidative stress and the increase in adhesion molecules, and the aggregation of platelets [1]. The rupture of the fibrous cap in an atherosclerotic plaque can result in the development of a life-threatening ASCVD, such as myocardial infarction and stroke, both of which can also lead to various complications that significantly impact the patient’s quality of life [6,7]. Furthermore, ASCVD is commonly asymptomatic up until the occurrence of acute coronary syndrome or stroke [8]. Therefore, preventive measures and early interventions aimed at reducing the risk of these conditions are crucial.

Type 2 diabetes mellitus (T2DM) is closely linked to an increased risk of CAD and cerebrovascular disease. Patients with T2DM have a 2–4-fold higher CVD risk compared to that of the general population, regardless of the presence/absence of other risk factors [9]. Thus, diabetes mellitus is regarded as being equivalent to the risk of CAD. Furthermore, ASCVD is the principal cause of death and disability among patients with diabetes mellitus [10]. Diabetes mellitus is characterized by insulin resistance, dyslipidemia, inflammation, and hypercoagulability, which exacerbate the underlying causes of atherosclerosis and heart failure [11]. To date, the cardiovascular benefit of multifactorial risk-reduction approaches using antidiabetic agents, antihypertensive drugs, and lipid-lowering therapy has been proven, thus reducing the incidence of cardiovascular complications [12]. However, the current therapeutic strategies are not sufficient for preventing adverse cardiovascular events, and a substantial number of patients with diabetes mellitus still face a risk of CVD, leading to the term “residual risk” [12].

For the management of the residual risk, international guidelines from the USA and Europe recommend the use of antiplatelet agents as a main management strategy in patients with diabetes and a high CVD risk [13,14]. Hence, antiplatelet agents such as aspirin, clopidogrel, and cilostazol could be contemplated in the management of the residual risk, pending additional supportive evidence.

Cilostazol is an antiplatelet agent that inhibits phosphodiesterase-3 (PDE-3) and increases cyclic adenosine monophosphate (cAMP) concentrations, leading to the inhibition of platelet aggregation and thrombus formation [15]. PDE-3 is expressed in cell populations such as platelets, VSMCs, cardiac myocytes, and adipocytes [16]. Therefore, cilostazol is expected to have some effects in these tissues.

Cilostazol has been approved for improving intermittent claudication symptoms in various countries. In further research, cilostazol treatment reduced the progression of atherosclerosis in patients with carotid artery stenosis [17,18] and the recurrence of stroke in patients with cerebrovascular disease [19,20]. In addition, previous studies have shown that cilostazol therapy is beneficial for CAD when administered immediately after percutaneous coronary intervention or in addition to other antiplatelet agents [21,22].

In 2015, a critical review of the literature regarding cilostazol and its role against CVD was reported [23]. Since then, many basic and clinical studies have been published; however, its role has not been reviewed systematically. Moreover, cilostazol has not been well documented in the literature with regard to its role in CVD and diabetes mellitus. In this review, we discuss the mechanism of action of cilostazol and evaluate the efficacy and safety of the drug based on the results of clinical trials. Furthermore, we comprehensively explore the role of cilostazol in the prevention and treatment of atherosclerosis.

## 2. Limitations of Aspirin Therapy for Atherosclerotic CVD

Aspirin, which inhibits COX-1 and prevents the formation of thromboxane, thereby exerting antiplatelet effects, remains the most commonly used antiplatelet agent [24]. Recently, controversy has arisen regarding the effectiveness of aspirin in the primary prevention of CVD. Both Asian and European studies that investigated the primary preventive effect of aspirin for CVD in patients with T2DM reported an absence of significant differences in the incidence of CVD between the aspirin treatment group and the non-aspirin group [25,26]. A meta-analysis of several clinical studies in which aspirin was administered for the primary prevention of CVD also showed that aspirin did not reduce the incidence of CVD in patients with diabetes [27]. The six-year follow-up of the primary preventive effect of aspirin against CVD in more than 14,000 Japanese elderly patients also failed to demonstrate the beneficial effect of aspirin on the incidence of CVD [28].

Moreover, it was reported that the use of aspirin not only increased the number of adverse events, such as major bleeding and gastrointestinal discomfort, but also did not reduce the risk of cardiovascular events in primary prevention (Table 1) [25,26,28,29,30,31,32,33,34].

There is some evidence from basic studies suggesting that aspirin might not be sufficient to reduce abnormal platelet activation. The platelet multidrug resistance protein 4 (MRP4) is upregulated after aspirin administration, which could represent one of the mechanisms underlying the high residual platelet reactivity after aspirin intake [35]. Importantly, patients with persistent high platelet reactivity while on aspirin therapy exhibited an increased risk of recurrent cardiovascular events [36].

Based on these findings, the role of aspirin in this context has changed drastically since 2018. The US Preventive Services Task Force did not recommend the initiation of low-dose aspirin among patients aged 60 years or older [37]. The American Heart Association (AHA) and American College of Cardiology (ACC) Foundation’s statement on aspirin for the primary prevention of cardiovascular events in individuals with diabetes suggests that low-dose aspirin should not be administered to those with a low risk for ASCVD (10-year risk < 5%), but can be considered in patients with an increased cardiovascular risk (10-year risk > 10%) [38,39]. Unfortunately, no direct evidence from a clinical trial supports these risk-based treatment recommendations, resulting in controversy in the primary prevention among patients with a high CVD risk. Moreover, the primary preventive effect of aspirin was not clearly demonstrated, even for high-risk patients, such as those with T2DM (Table 1).

In contrast, the guidelines for the secondary prevention of diabetes recommend aspirin therapy (75–162 mg daily) for individuals with diabetes and a history of ASCVD [13]. Nevertheless, concerns regarding the side effects of this therapy remain unaddressed. In the Aspirin in Reducing Events in the Elderly (ASPREE) study, cancer-related death occurred in 3.1% of the participants in the aspirin group and in 2.3% of those in the placebo group (hazard ratio: 1.31; 95% confidence interval (CI): 1.10–1.56), indicating a higher cancer mortality rate in the aspirin therapy compared with that of the placebo group [33]. Although the risk of cancer development was not confirmed in other studies (Table 1), concerns regarding this issue remain.

## 3. Beneficial Role of Cilostazol in the Development of Atherosclerosis

### 3.1. Antiplatelet Activity

Cilostazol inhibits platelet function by inhibiting PDE-3 and increasing cAMP levels [40] (Figure 1). Platelet reactivity is mediated by the intracellular calcium levels, which induce a molecular cascade including cAMP, resulting in platelet degranulation, conformational changes, and aggregation [41]. Elevated cAMP levels upregulate the activity of protein kinase A (PKA), which phosphorylates key intermediary molecules that are essential for platelet aggregation [42]. PKA has also been tied to the downregulation of platelet adhesion through the phosphorylation of the GP-Ib-IX complex, the inhibition of intracellular calcium store release, thromboxane receptor desensitization, and the inhibition of the phosphorylation of myosin light chain kinase [43].

Recent evidence has suggested a role for cilostazol in the inhibition of MRP4 [44] and in overcoming a high on-aspirin-treatment residual platelet reactivity in a cAMP-independent manner [45]. MRP4, which is expressed in platelets primarily in the membranes of dense granules [46], promotes platelet aggregation by transporting cyclic nucleotides, including cAMP and cGMP, outside of platelets [47]. MRP4 has also been shown to desensitize platelets to the inhibitory effects of endothelial-derived nitric oxide and sensitize them to ADP-mediated activation [44]. Thus, the inactivation of MRP4 and the inhibition of phosphodiesterase-4 by cilostazol are reported to have synergistic and potent therapeutic antiplatelet effects [46].

The clinical findings of decreased platelet reactivity after cilostazol treatment have been described in various populations, including patients with acute coronary syndrome and ischemic stroke [40,48], and even when administered in addition to aspirin and clopidogrel [49]. Cilostazol is likely to be more effective in inhibiting platelet activation, especially in patients with diabetes mellitus. Platelets themselves have insulin receptors which, when activated, initiate a molecular cascade, leading to increased cAMP levels and platelet inhibition [50]. Insulin resistance is associated with an attenuated P2Y1 response caused by low cAMP levels, which may explain the observation that patients with T2DM are at an increased risk of thrombosis [50]. Another study further established a relatively higher degree of platelet inhibition by cilostazol in patients with diabetes compared with those without this condition [51].

### 3.2. Increase in Nitric Oxide Secretion

Cilostazol has a potent vasodilatory effect. Cilostazol treatment leads to PKA phosphorylation and the activation of endothelial nitric oxide synthase (eNOS), which increases nitric oxide levels [52]. The vasodilatory effect of cilostazol therapy includes the PKA-mediated phosphorylation of G-protein-coupled receptor kinase 2, the inactivation of G-alpha-q-mediated signaling, and the hyperpolarization of smooth muscle cell membranes through the potentiation of calcium-activated potassium currents [53,54].

Because of its PDE-3-inhibitory function, cilostazol administration resulted in the accumulation of cAMP, thus inducing vasodilation [55]. This process was mediated by the cAMP-dependent secretion of nitrogen oxide (NO) by vascular endothelial cells [56]. The other signaling pathway is endothelium-independent and involves the direct effect of cilostazol on VSMCs. cAMP build-up in these cells induces the PKA-dependent activation of the calcium-dependent potassium channel, leading to vasorelaxation [57]. In addition, cilostazol was shown to inhibit cGMP-specific PDE or PDE-5 [58], which may contribute to its vasorelaxant properties by promoting the accumulation of cGMP in VSMCs [58]. Furthermore, cilostazol treatment attenuated adenosine uptake by several types of cells, thereby leading to adenosine build-up in the interstitial fluid and vascular circulation [59]. The increase in extracellular adenosine potentiates adenosine receptor A2 activity in endothelial cells, causing arterial dilatation [60].

### 3.3. Antiproliferative Activity, Including Alterations in Adhesion Molecules

Cilostazol exhibited an antiproliferative effect on the vasculature by reducing neointima formation in animal models [61,62]. This antiproliferative effect of cilostazol is likely mediated by its effects on cAMP elevation and PKA inhibition, which are responsible for the decrease in the transcription of genes encoding cellular adhesion molecules and the proliferation of VSMCs [63] (Figure 1). The administration of cilostazol also decreased the levels of the vascular cell adhesion molecule, intercellular adhesion molecule, and E-selectin [64,65]. A decrease in the expression of these cellular adhesion molecules disrupts neointimal proliferation through the reduction in inflammatory cell invasion into the damaged intima [66]. In addition, cilostazol therapy downregulates growth factors, such as the vascular endothelial growth factor and platelet-derived growth factor [62,65]. Furthermore, cilostazol plays a favorable role in the transition to smooth muscle cells from a de-differentiated state [67].

### 3.4. Improvement in Vascular Endothelial Function

Cilostazol has beneficial effects on endothelial function, which is linked to NO production (Figure 1). Cilostazol treatment increased cAMP levels in endothelial cells, which led to NO production [68]. In aged Wistar rats, endothelial dysfunction was restored by cilostazol treatment [69]. This protective effect of cilostazol could be attributed to its favorable actions on arterial walls, including a reduction in oxidative stress and an increase in NO bioavailability [69]. Another study proved that cilostazol treatment decreased intimal thickening and improved the vasodilation capacity [70]. Cilostazol stimulates the production of hepatocyte growth factor, which is known for its favorable role in attenuating the malformation and apoptosis of endothelial cells [71].

Also, in a clinical setting, a three-month intervention using cilostazol improved endothelial function, as measured by flow-mediated dilation, compared with aspirin in patients with stroke (1.0% vs. 0.8%) [72]. Moreover, a six-month treatment with cilostazol improved vasomotor reactivity, as evidenced by an increase in the coronary luminal diameter following the administration of N-monomethyl-L-arginine [73].

### 3.5. Anti-Inflammatory Effect

In an animal study, cilostazol deactivated the nucleotide-binding domain-like receptor protein 3 (NLRP3) inflammasome [74], which is associated with atherosclerosis and CVD [75,76]. This anti-inflammatory potential was mediated by the downregulation of p65 NF-κB and the enhancement of sirtuin-1, leading to the downregulation of NLRP3, an apoptosis-associated speck-like protein, active caspase-1, and interleukin-1β [74]. In an animal study of hypercholesterolemic rats, cilostazol treatment protected against hypercholesterolemia-induced cardiac damage via molecular mechanisms targeting the crosstalk between Nrf2 induction and NF-κB inhibition in the heart [77]. Another study using the mouse models of myocardial ischemia–reperfusion showed that cilostazol treatment attenuated multiple inflammatory markers through the activation of the peroxisome proliferator-activated receptor-γ (PPARγ), Janus kinase 2 (JAK2), and the signal transducer and activator of transcription 3 (STAT3) pathways [78].

Similar findings were obtained in human studies. In a randomized controlled trial (RCT), treatment with cilostazol (100 mg twice daily) significantly decreased the hsCRP levels by 23.6%, erythrocyte sedimentation rate by 38.7%, and plasma malondialdehyde concentration by 17.6% [79]. Our group also found that treatment with cilostazol (200 mg once daily) decreased hsCRP levels significantly compared with aspirin in patients with T2DM and coronary artery stenosis [80]. It is suggested that the greater anti-inflammatory potency of cilostazol, when added to the standard dual antiplatelet therapy of aspirin and clopidogrel, may have contributed to the inhibition of the progression of carotid intima–media thickness, without raising the risk of hemorrhage [81]. These data are indicative of the deactivation of inflammation by cilostazol therapy.

### 3.6. Antioxidative Stress Effect

Cilostazol treatment is related to the activation of endothelial nitric oxide synthase (eNOS), which is implicated in the maintenance of endothelial function and the reduction in superoxide-radical-induced injury [52]. Cilostazol upregulates eNOS activity through multiple pathways. The inhibition of adenosine reuptake increases the activation of an adenosine-mediated signaling mechanisms involving the “survival” kinase Akt, which subsequently activates eNOS via phosphorylation [52]. The cilostazol-mediated inhibition of the phosphatase and tensin homolog also upregulates phosphotadidylinositol-3-phosphate, which in turn activates Akt. The cAMP-mediated activation of PKA has also been shown to activate eNOS through phosphorylation at a site distinct from Akt [52].

Through these basic mechanisms, cilostazol has been shown to attenuate ischemia–reperfusion injury in models of myocardial, hepatic, and spinal cord reperfusion injury [82,83]. The administration of cilostazol prior to coronary occlusion and reperfusion was associated with a significantly smaller infarct size in a rabbit model compared with that of the controls [84].

Oxidative stress was also decreased in a clinical trial involving individuals with T2DM and hypertension [79]. Plasma malondialdehyde and blood glutathione markers were improved after a one-month treatment with cilostazol, indicating its potential antioxidative effects in humans [79].

### 3.7. Improvement in the Lipid Profile

Cilostazol seems to induce this favorable alteration in lipid metabolism through the upregulation of lipoprotein lipase in adipose tissue [85]. The administration of cilostazol increased lipolysis by inhibiting PDE-3 in adipocytes [86] (Figure 1). In an animal study, the administration of cilostazol enhanced LDL receptor-related protein 1 (LRP1) expression in the liver by activating PPARγ through the peroxisome proliferator-responsive elements present in the LRP1 promoter [87]. This upregulation of hepatic LRP1 may play an important role in improving HDL-cholesterol and triglycerides [88]. In the studies of animals fed with a high-cholesterol diet, the administration of cilostazol decreased the triglyceride and cholesterol content in the vascular system [89,90].

It Is noteworthy that the Increase In cAMP triggered by cilostazol treatment affects norepinephrine, thus increasing fat metabolism and exothermic reactions, and consequently increasing energy consumption. Cilostazol can increase uncoupling protein 1 (UCP1) in adipocytes [91]. Regulating the expression of UCP1 and PGC1α through cAMP-responsive element binding (CREB) and increasing free fatty acid release through lipolysis are used as fuel for thermogenesis in mitochondria [92].

Several clinical trials that examined the effects of cilostazol on lipid profiles have reported similar findings [80,93,94]. In these studies, the triglyceride levels decreased by 15–34% in patients with PAD or T2DM. In contrast, HDL-cholesterol was reported to increase by 7–20% [18]. Cilostazol therapy reduced the remnant lipoprotein level, which is associated with an increased risk of CVD [94]. Furthermore, cilostazol treatment reduced apolipoprotein B levels, which are an indirect indicator of LDL particle number [95]. These improvements in lipid profiles afforded by cilostazol therapy help prevent the development and progression of atherosclerosis.

## 4. Clinical Results of Cilostazol Treatment for Preventing Cardiovascular Events

### 4.1. Methods for Meta-Analysis of Major Adverse Cardiovascular Outcomes

For clinical interpretation, we systematically reviewed the effects of cilostazol on cardiovascular outcomes, as registered in PROSPERO CRD42023444959. We searched PubMed, Embase, and the Cochrane databases for English-language studies published up to July 2023. The search terms included keywords such as ‘cilostazol’ for the study intervention; ‘major adverse cardiovascular event’, ‘cardiovascular disease’, ‘cardiovascular event’, ‘myocardial infarction’, and ‘stroke’ for the outcomes; and ‘randomized controlled trial’ for the study design. The full list of search terms can be found in Appendix A. Using this process, we selected the studies to be included in the present analysis based on the following inclusion criteria: studies that (1) compared cilostazol with placebo, control, aspirin, or clopidogrel; (2) reported composite major adverse cardiovascular events (MACEs); and (3) were RCTs. We excluded studies with a follow-up duration of less than 3 months.

The primary outcome was MACE. Secondary outcomes Included individual cardiovascular events, such as fatal or non-fatal MI, fatal or non-fatal stroke, cardiovascular death, and all-cause mortality. Other adverse events, including major bleeding, discontinuation, headache, and palpitation, were also analyzed.

Study selection and data collection were confirmed by both authors. Logarithmic hazard ratios (HRs) were used to calculate combined estimates with a random-effects model. Random-effects risk ratios were calculated for studies that provided only the event numbers. The included trials were assessed for the risk of bias using the Cochrane Risk of Bias 2 tool (see Appendix A). The *I^2^* statistic was employed to evaluate the overall heterogeneity of all comparisons. Publication bias was assessed using a funnel plot and Egger’s test. Sensitivity analyses were performed focusing on the regions of the trials. The meta-analysis was conducted using the ‘metafor’ statistical package in R software (version 4.1.0; R Development Core Team, Vienna, Austria).

### 4.2. Meta-Analysis Results of Major Adverse Cardiovascular Outcomes

Out of 496 articles, 34 were selected [18,20,95,96,97,98,99,100,101,102,103,104,105,106,107,108,109,110,111,112,113,114,115,116,117,118,119,120,121,122,123,124,125] (Figure 2), with five trials targeting patients with diabetes (patients with diabetes included, ranging from 8.8% to 56.0% in non-targeted trials) (Table 2). A total of 29 studies were conducted for secondary prevention among patients suffering from CAD, stroke, or PAD. Cilostazol was concurrently used with aspirin in 19 studies, and as part of a triple regimen including aspirin and clopidogrel in 13 studies. The bias of the meta-analyses is reported in Appendix A.

All 33 studies mentioned above were included in the meta-analysis of cilostazol therapy for the prevention of composite MACE. The definition of MACE varied among the trials, which might be attributed to differences in patient characteristics (Appendix A). Cilostazol administration effectively reduced the risk of MACE compared with both the control (risk ratio (RR): 0.62; 95% CI: 0.52–0.73) (Figure 3A) and aspirin (RR: 0.74; 95% CI: 0.63–0.87) treatments (Figure 3B). Cilostazol exhibited a superior efficacy over aspirin for secondary prevention, whereas further evidence is needed to draw a solid conclusion regarding its efficacy in primary prevention (Figure 3B). Our results confirmed the favorable effects of cilostazol on stroke, which are even better than those of aspirin [126] (Figure 4B). The beneficial efficacy of cilostazol in reducing the risk of myocardial infarction and cardiovascular death was not demonstrated (Figure 4A,C). The distinction in responses between ischemic stroke and myocardial infarction might lie in their differing associations with underlying etiologies [127]. Ischemic stroke is primarily associated with the clogging of blood vessels. In contrast, in myocardial infarction, multiple blood vessels may be involved, and other risk factors often play a significant role before an attack occurs.

Cilostazol therapy yielded a reduced bleeding risk compared with aspirin (Figure 4E). However, its usage was limited because of adverse events, leading to therapy discontinuation (Figure 4F), such as headache and palpitation (Appendix A). This suggests the need for improvements in this medication to address its side effects.

### 4.3. Coronary Artery Disease (CAD)

To date, the effect of cilostazol on CAD has not been fully elucidated, particularly as a primary prevention. In the ESCAPE study, we investigated the efficacy and safety of cilostazol and aspirin as the primary prevention of CAD in patients with T2DM. This investigation comprehensively evaluated CAD by assessing coronary artery stenosis, plaque composition, and calcium deposition using up-to-date computed tomography angiography [80]. A sustained-release form of cilostazol treatment (200 mg/day) for 12 months reduced the progression of coronary artery stenosis in patients with diabetes compared with aspirin (−3.1% vs. 1.7%). Notably, the noncalcified plaque component was the main portion of atheromatous plaques that was reduced by cilostazol treatment [80].

We extended this observation to investigate the long-term effect of cilostazol and aspirin (ESCAPE-extension study) [123]. In this extension study, we found that cilostazol treatment reduced the incidence of cardiovascular events in the patients with T2DM compared with aspirin over a median follow-up of 5.2 years (hazard ratio: 0.24; 95% CI: 0.07–0.83). This benefit of cilostazol therapy was maintained together with age, sex, blood pressure, LDL-cholesterol, and coronary artery calcium score, without serious adverse events [123].

A previous study that compared cilostazol with aspirin for primary prevention in patients with T2DM failed to observe favorable effects for cilostazol [118]. The overall cardiovascular event rate was very small (<3% in both the aspirin and cilostazol groups), which may have hampered the detection of any significant differences in this population. The overall data suggest the potential beneficial role of cilostazol treatment in the primary prevention of CVD, although additional evidence is still needed.

### 4.4. Carotid Artery Stenosis and Cerebrovascular Disease

Cilostazol decreased the carotid intima–media thickness (IMT), which is a surrogate indicator of CVD. In the Diabetic Atherosclerosis Prevention by Cilostazol study, which was conducted in patients with T2DM accompanied by arteriosclerosis obliterans, the carotid IMT was increased in the aspirin group (+0.048 mm), whereas the cilostazol group exhibited a decrease in this parameter (–0.024 mm), resulting in a statistically significant difference in carotid IMT within 2 years [18].

The Cilostazol Stroke Prevention Study compared the secondary stroke preventive effects by dividing patients with a history of stroke into a cilostazol group and an aspirin group. As a result, the frequency of stroke recurrence was about 26% lower in the cilostazol group vs. the aspirin group [20]. Moreover, dual antiplatelet therapy using cilostazol together with aspirin or clopidogrel reduced the incidence of ischemic stroke recurrence in patients at high risk of this condition [119]. In that study, the risk of severe or life-threatening bleeding was similar between the cilostazol with aspirin or clopidogrel therapy and the aspirin or clopidogrel alone therapy.

In a network meta-analysis comparing 11 antiplatelet medications, cilostazol monotherapy yielded the most favorable results in stroke recurrence and bleeding risk over other agents in the patients who had experienced a stroke or transient ischemic attack (TIA) [128]. Further studies are needed for directly comparing cilostazol with a therapeutic agent other than aspirin.
ijms-25-02593-t002_Table 2Table 2Randomized controlled studies of cilostazol for preventing cardiovascular outcomes.Bibliographic InformationPopulationIntervention *Duration (Intervention/Observation)Patients’ NumberMen (%)/Age (Years)Diabetes Gotoh F. et al., 2000 [95]Patients who had cerebral infarction within the previous 1–6 monthscilostazol 200 mg/day vs. placebo1 year/3 years105265.6/65.124.7%CREST study [96]Patients with angina or silent myocardial ischemia undergoing coronary stent implantationcilostazol 200 mg/day + clopidogrel + aspirin vs. placebo + aspirin + clopidogrel6 months/6 months526 74.2/6025.7%Chen Y.D. et al., 2006 [97]Patients undergoing bare metal stent implantationcilostazol 200 mg/day + clopidogrel + aspirin vs. placebo + aspirin + clopidogrel6 months/9 month10661.7/5829.2%DECLARE-LONG study [98]Patients with angina pectoris and/or positive stress testcilostazol 200 mg/day + clopidogrel + aspirin vs. aspirin + clopidogrel6 months/9 month500 64.2/61.133.2%CASTLE study [99]Patients with clinical diagnosis of PAD and symptoms of claudicationcilostazol 200 mg/day (or 100 mg/day) vs. placebo3 years/3 years143565.5/66.235.7%CASISP study [100]Patients who had ischemic stroke within the previous 1–6 monthscilostazol 200 mg/day vs. aspirin 100 mg/day12–18 months/18 months719 68.7/60.218.2%DECLARE-DIABETES study [101]Patients with diabetes and angina pectoris or positive stress test and a native coronary lesioncilostazol 200 mg/day + clopidogrel + aspirin vs. aspirin + clopidogrel6 months/9 month40058/60.9100%Guo J.J. et al., 2009 [102]Patients who had ischemic stroke during the recent 1–6 monthscilostazol 200 mg/day vs. aspirin 100 mg1 year/1 year6835.3/60.88.8%Han Y. et al., 2009 [103]Patients with acute coronary syndromes undergoing PCIcilostazol 200 mg/day + clopidogrel + aspirin vs. aspirin + clopidogrel6 months/1 year121273.3/59.921.7%Soga Y. et al., 2009 [104]Patients with intermittent claudication undergoing endovascular therapycilostazol 200 mg/day + aspirin 81 or 100 mg/day vs. aspirin 81 mg or 100 mg/day2 years/2 years7883.3/70.735.9%DAPC study [18]Patients with T2D suspected of PAD cilostazol 100 mg or 200 mg/day vs. aspirin 81 mg or 100 mg/day 2 years/2 years29752.2/63.5100%CSPS 2 study [20]Patients who had cerebral infarction within the previous 26 weekscilostazol 200 mg/day vs. aspirin 81 mg/day1–5 years/5 years267271.7/63.529.0%TOSS-2 study [105]Patients with acute symptomatic stenosis in the middle cerebral artery or the basilar arterycilostazol 200 mg/day vs. clopidogrel7 months/7 months45751.2/65.542.5%DECLARE-LONG II study [106]Patients with stable angina or acute coronary syndrome and a native long coronary lesioncilostazol 200 mg/day + clopidogrel + aspirin vs. placebo + aspirin + clopidogrel8 months/1 year49970.6/61.535.3%CAIST study [107]Patients who had ischemic stroke with neurological deficitscilostazol 200 mg/day vs. aspirin 300 mg/day90 days/90 days45861.4/6334.7%CILON-T study [108]Patients undergoing drug-eluting stentcilostazol 200 mg/day + clopidogrel + aspirin vs. aspirin + clopidogrel6 months/6 months91570.7/64.433.6%STOP-IC study [109]Patients with symptomatic PAD cilostazol 200 mg/day + aspirin vs. aspirin12 months/12 months19168.6/7356.0%Gao W. et al., 2013 [110]Obese patients undergoing coronary stentingcilostazol 200 mg/day + clopidogrel + aspirin vs. aspirin + clopidogrel1 year/1 year42880.4/56.517.8%Youn Y.J. et al., 2014 [111]Patients with CAD implanting drug-eluting stentcilostazol 200 mg/day + clopidogrel + aspirin vs. aspirin + clopidogrel3 months/1 year61563.6/64.631.7%CATHARSIS study [112]Patients with intracranial arterial stenosiscilostazol 200 mg/day + aspirin vs. aspirin2 years/2 years16365.6/68.336.8%CATS-I study [113]Patients with stable CAD or acute coronary syndrome cilostazol 200 mg/day + aspirin vs. aspirin2 year/2 year51474.7/6939.9%Zheng X.-T. et al., 2016 [114]Patients with complex lesions undergoing PCIcilostazol 100 mg/day + clopidogrel + aspirin vs. aspirin + clopidogrel3 months/1 year12770.9/64.637.0%CREATIVE study [115]Patients undergoing PCIcilostazol 200 mg/day + aspirin + clopidogrel vs. aspirin + clopidogrel (75 mg or 150 mg)12 months/18 months107659.9/58.433.2%DECREASE PCI study [116]Patients undergoing drug-eluting stents for CAD cilostazol 200 mg/day + aspirin + clopidogrel vs. placebo + aspirin + clopidogrel1 year/1 year40471.3/62.230.9%PICASSO study [117]Patients with ischemic stroke with intracerebral hemorrhage or multiple microbleedscilostazol 200 mg/day vs. aspirinAt least 1 year/median 1.9 years151262/65.732.1%Hong S et al., 2019 [118]Patients with T2D without macrovascular complicationscilostazol 200 mg/day vs. aspirin3 years/3 years41550.4/59.6100%CSPS.com study [119]Patients with high-risk non-cardioembolic ischemic strokecilostazol 200 mg/day + aspirin or clopidogrel vs. aspirin or clopidogrelAt least 1 year/median 1.4 years187970.3/69.737.3%Chen Y.-C. et al., 2019 [120]Patients undergoing PCIcilostazol 200 mg/day + clopidogrel + aspirin vs. aspirin + clopidogrel6 months/2 years21169.7/65.725%Uchiyama S. et al., 2021 [121]Patients with intracranial arterial stenosiscilostazol 200 mg/day + aspirin or clopidogrel vs. aspirin or clopidogrelAt least 1 year/median 1.4 years54767.1/7039.5%Kalantzi K. et al., 2021 [122]Patients with T2D exhibiting symptomatic lower extremity arterial diseasecilostazol 200 mg/day + clopidogrel vs. clopidogrel12 months/3 years79464/67.8100%ESCAPE-Extension: Sohn M. et al., 2022 [123]Patients with T2D and mild to moderate coronary artery stenosiscilostazol 200 mg/day vs. aspirinAt least 1 year/median 5.2 years8863/61100%Lin J.-L. et al., 2022 [124]Patients with CAD or at a high risk of cardiovascular diseasecilostazol 200 mg/day vs. placebomean 2.9 years/mean 2.9 years26673.3/65.839.5%LACI-2 study [125]Patients who had lacunar strokecilostazol 200 mg/day ± isosorbide mononitrate vs. isosorbide mononitrate or no study drug12 months/12 months36369.1/64.322.0%Park S. et al., 2023 [129]Patients who had ST-elevation myocardial infarctioncilostazol 200 mg/day + aspirin or clopidogrel vs. aspirin or clopidogrel1 or 6 months/12 months94881.5/61.626.6%* The standard dosage of other interventions was not specified: aspirin, 100 mg, and clopidogrel, 75 mg. CAD, coronary artery disease; PAD, peripheral arterial disease; PCI, percutaneous coronary intervention; and T2D, type 2 diabetes.
Figure 3Forest plots of cilostazol in composite major adverse cardiovascular events (MACEs) compared with the following: (**A**) the control, (**B**) aspirin, and (**C**) clopidogrel. CI, confidence interval; RR, risk ratio.
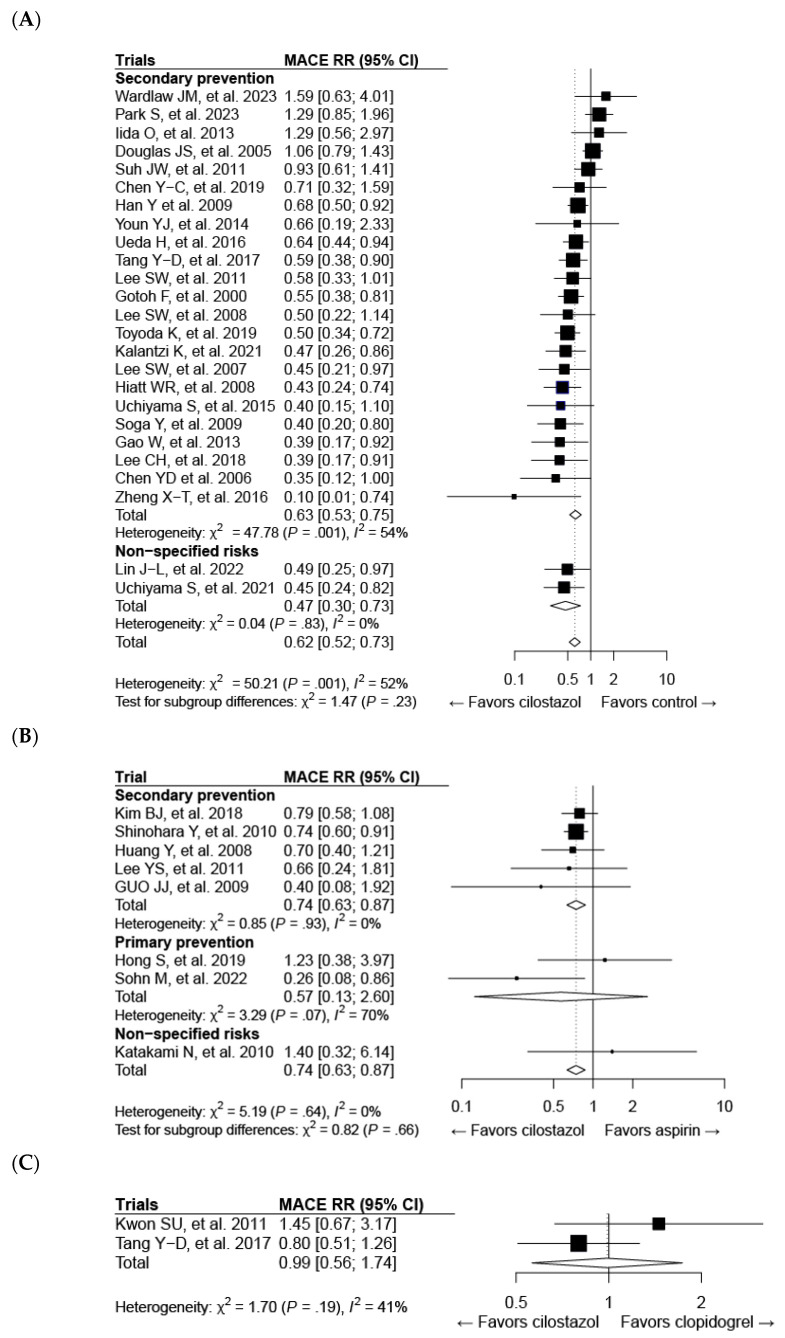

Figure 4Forest plots of cilostazol in cardiovascular and safety events: (**A**) myocardial infarction, (**B**) stroke, (**C**) cardiovascular death, (**D**) all-cause death, (**E**) bleeding, and (**F**) discontinuation of the study drug.
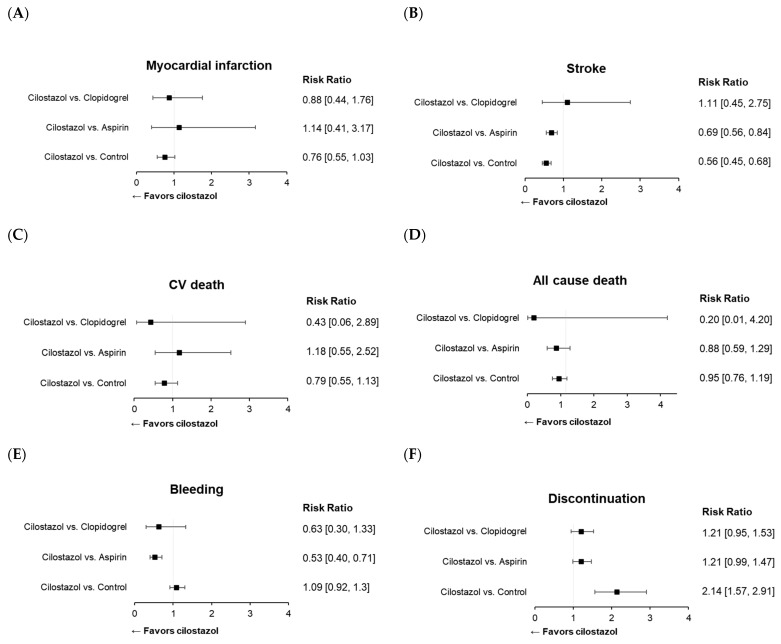


### 4.5. Peripheral Artery Disease (PAD)

The clinical excellence of cilostazol is sufficient to recommend it as a first-line drug in PAD, because it increases NO in vascular endothelial cells in a dose-dependent manner, to dilate blood vessels [130]. A meta-analysis of 16 RCTs reported that intermittent claudication was significantly improved by cilostazol therapy (100–300 mg per day) compared with the placebo [131]. An RCT including 698 patients with moderate-to-severe claudication symptoms reported that a 6-month cilostazol therapy improved the maximal walking distance by 54%, compared with pentoxifylline therapy (30% increase) [132]. Another study including 394 patients with intermittent claudication showed that cilostazol treatment for 6 months improved the maximal walking distance compared with the placebo [133]. A 2021 Cochrane Systematic Review and Meta-analysis including 16 double-blind RCTs reported that participants taking cilostazol had a higher initial claudication distance, by 26.5 m (95% CI: 18.9–34.1 m), compared with those taking the placebo [131]. Cilostazol exerted beneficial effects regarding the primary prevention of diabetic foot ulcers via a lowering effect on MMP-9 levels [134].

Recently, a prospective study with 794 participants showed that the adjunctive administration of cilostazol to clopidogrel-treated patients with T2DM with symptomatic PAD lowered the risk of ischemic events and improved intermittent claudication symptoms, without increasing the bleeding risk [122]. Based on this evidence, the ACC/AHA guidelines provided a class I recommendation for the use of cilostazol in the treatment of intermittent claudication in 2016 and thereafter [135].

## 5. Potential Benefits of Cilostazol Therapy

### Neuroprotective Effect

There is evidence supporting the benefits of cilostazol in neuroprotection. Several studies reported that cilostazol was helpful in preventing symptomatic cerebral vasospasm and in reducing cerebral infarction and intracranial hemorrhage in patients with stroke [136,137]. These effects depended mainly on the anti-inflammatory and antiapoptotic effects of cilostazol, which are mediated by scavenging hydroxyl radicals, decreasing the formation of tumor necrosis factor-α (TNF-α), and inhibiting adenosine diphosphate-ribose polymerase activity [138]. Moreover, cilostazol treatment improved neuronal survival by promoting the Akt and CREB, as well as brain-derived neurotrophic factor content in a rat model [74]. The production of NO triggered by cilostazol therapy via the activation of endothelial NO synthase also contributed to neuroprotection via endothelial protection [139].

It is known that cilostazol reduces blood–brain barrier permeability and the degree of intracerebral cell death via mechanisms involving the inhibition of collagenase [140]. Cilostazol treatment also prevented endothelial cell death and protected collagen type 4, laminin, and vascular endothelial and N-cadherins from collagenase injury, and attenuated blood–brain barrier leakage, resulting in intracranial hemorrhage [141].

In addition, cilostazol had a neuroprotective effect against hypoperfusion and permanent cerebral ischemia [142,143]. Cilostazol treatment in Wistar rats with bilateral carotid artery occlusion decreased apoptosis by downregulating TNF-α, suppressed overly activated astroglia and microglia in the white matter, and increased oligodendrocytes [142]. The protection mechanism was multifactorial in reducing oxidative stress [143].

In a multicenter and double-blind RCT, the administration of cilostazol to participants with moderate or severe white matter changes (WMC), which demonstrates small-vessel disease in the brain, and at least one lacunar infarction significantly reduced the risk of ischemic vascular events compared with aspirin (HR: 0.11; 95% CI: 0.02–0.89) [144]. However, there was no significant difference between the effects of cilostazol and those of aspirin on WMC progression in patients with cerebral small-vessel disease [144]. In a systematic review and meta-analysis of RCTs for stroke and cognitive decline, cilostazol therapy reduced recurrent ischemic stroke; however, the authors failed to prove its efficacy on cognitive dysfunction [126].

## 6. Considerations in Using Cilostazol Therapy

### 6.1. Adverse Effects of Cilostazol Therapy

Cilostazol has been reported to cause more headaches, palpitations, and gastrointestinal disorders when compared with a placebo [145]. Headaches were the primary reason for the discontinuation of this medication. Cilostazol significantly increased the cerebral blood flow in patients with chronic cerebral infarction [95]; therefore, the cerebral blood vessels dilated, which may have caused headaches, although the patients would acclimate to it within 1 week. Currently, a sustained-release formulation that reduces the incidence of headache by introducing a technology with controlled dissolution is available in several countries [146].

It is important to understand that cilostazol has the potential to increase the heart rate by about 3–5 beats/min [23,145]. Although this side effect will not necessarily be present in all patients, and the extent of its impact may vary according to the dosage and individual physiological responses, further research is needed to investigate how this outcome may affect the progression of CVDs. Thus, cilostazol is contraindicated in patients with established heart failure [147].

### 6.2. Effects of Cilostazol Therapy by Regional Background

Although numerous studies have been published on cilostazol’s protective effects against atherosclerosis, it is evident that most of these studies have been conducted in Asia (Figure 5). Studies have demonstrated that cilostazol improves major adverse cardiovascular events (MACE) and stroke outcomes, affirming its efficacy. However, limited studies make it hard to interpret its effects in other regions (Figure 5A,C). Moreover, as recent studies have not shown favorable results for myocardial infarction [125,129], further investigation is needed to determine its efficacy in Asian populations as well (Figure 5B).

## 7. Conclusions

Cilostazol has multifaceted beneficial effects through various mechanisms, including platelet inhibition, vasodilation via NO production, protective effects on vascular endothelial cells, antiproliferative effects on VSMCs, and favorable changes in lipid profiles (Figure 6). These beneficial effects of cilostazol are attributable to its pleiotropic properties, including its anti-inflammatory and antioxidative actions. A neuroprotective effect was also found in a few clinical studies of cilostazol.

Based on the recent results of large clinical studies suggesting that aspirin should not be recommended in primary prevention settings, cilostazol treatment may be a viable option for preventing the development and progression of vascular diseases. Obviously, cilostazol is effective in PAD, and recent evidence supports its effectiveness in CAD as well as ischemic stroke. Notably, recent studies reported that cilostazol treatment was efficacious in reducing atheromatous plaque progression in coronary arteries and decreasing cardiovascular events compared with aspirin in patients with diabetes and subclinical CAD. Thus, cilostazol might be helpful in the comprehensive vascular management of patients with a high risk of CV. However, it is noteworthy that these results were mainly derived from East-Asian populations. Additional trials including other ethnic groups are needed to generalize the beneficial effects of this drug. Adverse events occurring after cilostazol therapy, such as headache and heart rate increase, have not been completely resolved; therefore, the development of more-advanced PDE-3 inhibitors that do not exhibit these unwanted effects is warranted. Longer-term studies with a robust outcome are needed to confirm the efficacy and safety of cilostazol therapy.

## Figures and Tables

**Figure 1 ijms-25-02593-f001:**
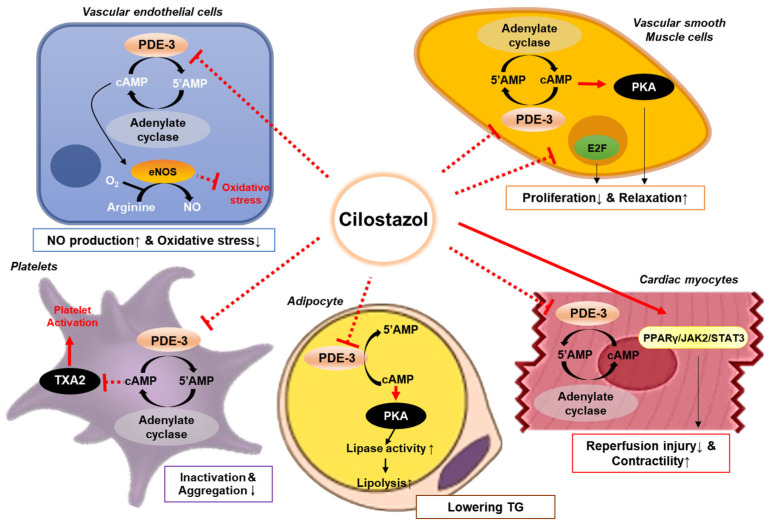
Mechanisms of action of cilostazol. Cilostazol inhibits phosphodiesterase-3 in various cell types including platelets, vascular cells, myocytes, and adipocytes, which affects several pathways resulting in vasodilation and enhanced blood perfusion. Cilostazol induces nitric oxide (NO) release from endothelial cells and activates calcium-dependent potassium channels in vascular smooth muscle cells, leading to vasodilation. This action leads to the upregulation of the protein kinase A (PKA), and eventually causing arterial dilatation and platelet inactivation. Furthermore, cilostazol attenuates the activation of platelets, thus facilitating blood perfusion. Abbreviations: 5′AMP, 5′-adenosine monophosphate; cAMP, cyclic adenosine monophosphate; eNOS, endothelial nitric oxide synthase; JAK2, Janus kinase 2; PDE-3, phosphodiesterase-3; PKA, protein kinase A; PPARγ, peroxisome proliferator-activated receptor gamma; STAT3, signal transducer and activator of transcription 3; TG, triglycerides; and TXA2, thromboxane A2.

**Figure 2 ijms-25-02593-f002:**
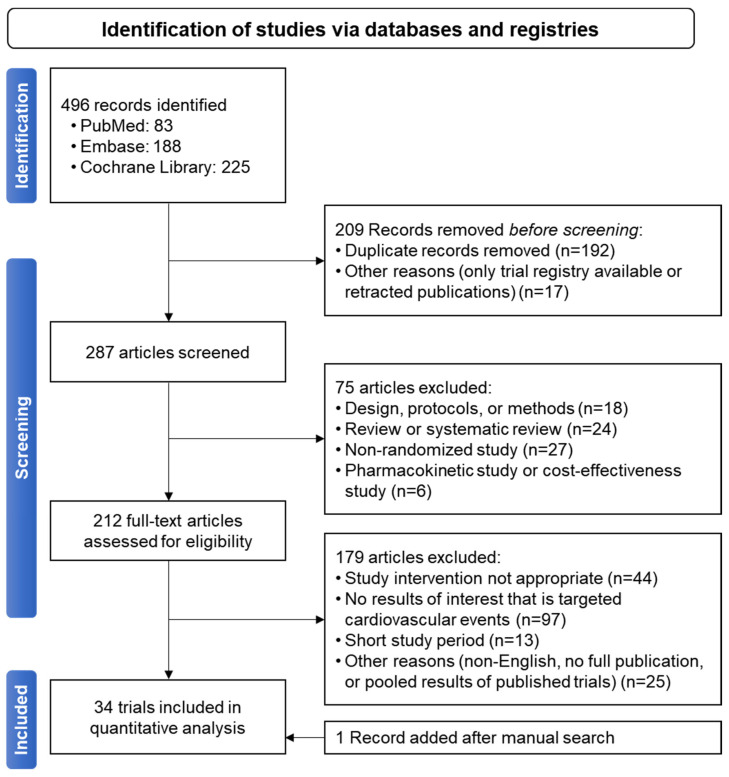
PRISMA flow diagram for studies included in the current meta-analysis.

**Figure 5 ijms-25-02593-f005:**
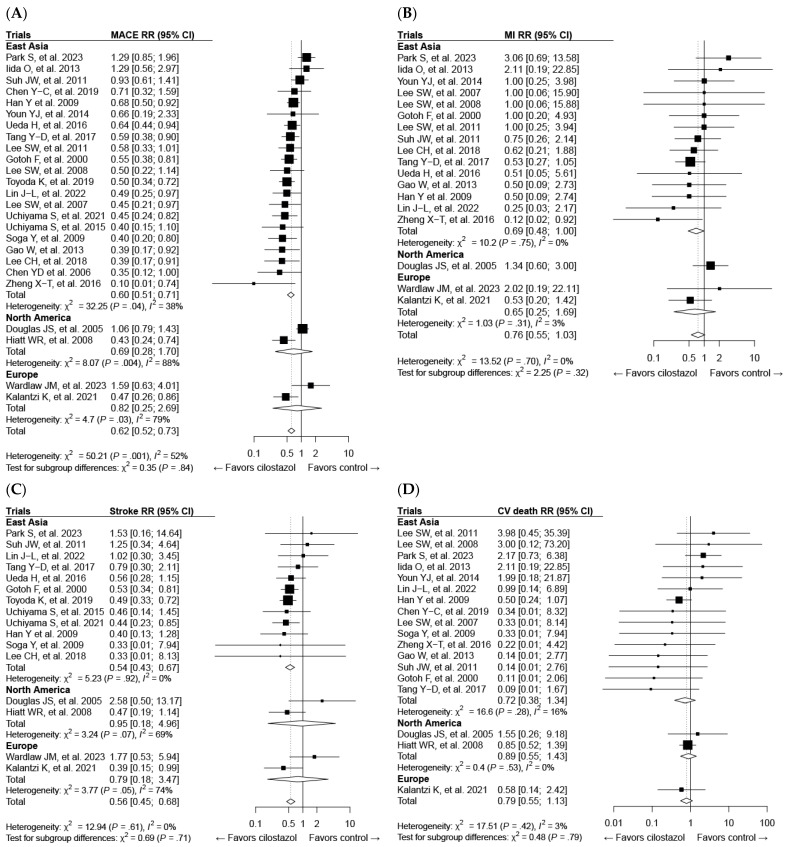
Forest plots of cilostazol’s effect on composite major adverse cardiovascular events (MACEs) and individual components categorized by regional background: (**A**) MACE, (**B**) myocardial infarction (MI), (**C**) stroke, and (**D**) cardiovascular death. CI, confidence interval; RR, risk ratio.

**Figure 6 ijms-25-02593-f006:**
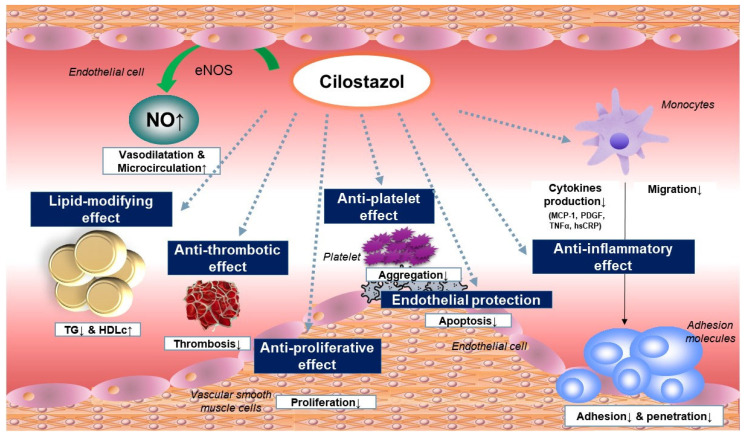
Anti-atherosclerotic effect of cilostazol.

**Table 1 ijms-25-02593-t001:** Randomized controlled studies with aspirin for preventing primary cardiovascular events since 2000.

Bibliographic Information	Trial Information *				Patient’s Characteristics	Efficacy and Safety Outcomes		
	Population	Duration **	Patients’ Number	Men (%)/Age (Years)	Diabetes Comorbidity	Cardiovascular Events ***	Bleeding Risks	Cancer Risks
PPP study [29]	Participants with at least one already known major cardiovascular risk factor	5 years	4495	42.5/64.4	16.5%	RR of 3P-MACE ^a^ with aspirin vs. no-aspirin: 0.71 (0.48–1.04)	Severe bleedings more frequent in the aspirin group (1.1% vs. 0.3%; *p* < 0.0008)	No difference in cancer risks with aspirin vs. no-aspirin (3.9% vs. 3.5%)
WHS study [30]	Female health professionals, aged ≥45 and without a history of cardiovascular disease or cancer	mean 10.1 years	39,876	0/54.6	2.6%	RR of 3P-MACE ^a^ with aspirin vs. placebo: 0.91 (0.80–1.03)	Any gastrointestinal bleeding: RR 1.22 (1.10–1.34) Requiring transfusion: RR 1.40 (1.07–1.83)	Not reported
JPAD study [25]	Patients with type 2 diabetes	mean 4.37 years	2539	54.6/64	100%	HR of atherosclerotic events ^b^ with aspirin vs. no-aspirin: 0.80 (0.58–1.10)	No difference in significant gastrointestinal bleeding with aspirin vs. no-aspirin (1.0% vs. 0.3%)	Not reported
POPADAD study [26]	Participants aged ≥40 with type 1 or type 2 diabetes and an ankle brachial pressure index of 0.99 or less but no symptomatic cardiovascular disease	median 6.7 years (4.5~8.6 years)	1276	44.1/60.3	100%	No difference in atherosclerotic events ^c^ with aspirin vs. no-aspirin: 0.98 (0.76–1.26)	No difference in gastrointestinal bleeding with aspirin vs. no-aspirin: 0.90 (0.53–1.52)	No difference in malignancy with aspirin vs. placebo (8.3% vs. 10.7%)
JPPP study [28]	Participants aged 60 to 85 years, presenting with hypertension, dyslipidemia, or diabetes mellitus without cardiovascular disease	6 years (median 5.02 years)	14,658	42.3/70.5	33.9%	HR of 3P-MACE ^a^ with aspirin vs. no-aspirin: 0.94 (0.77–1.15)	Major gastrointestinal and extracranial bleedings more frequent in the aspirin group (0.1% vs. 0.07%; *p* < 0.001)	Not reported
ASCEND study [31]	Patients with type 2 diabetes without cardiovascular disease	at least 7 years (mean 7.4 years)	15,480	62.6/63.3	100%	RR of serious vascular events ^d^ with aspirin vs. placebo: 0.88 (0.79–0.97)	RR of any major bleeding: 1.29 (1.09–1.52)	RR of any cancer: 1.01 (0.92–1.11)
ARRIVE study [32]	Participants aged ≥55 years (men) or ≥60 years (women) and had moderate cardiovascular risk, without at high risk of bleeding or diabetes	6 years (mean 5 years)	12,546	70.4/63.9	0%	HR of 3P-MACE ^a^ with aspirin vs. no-aspirin: 0.79 (0.61–1.02)	HR of gastrointestinal bleeding: 2.11 (1.39–3.28)	No difference in cancer with aspirin vs. placebo (not sufficient events)
ASPREE study [33]	Elderly participants who did not have cardiovascular disease, dementia, or disability	6 years (median 4.7 years)	19,114	43.6/median 74	10.8	HR of ischemic events-related deaths with aspirin vs. placebo: 0.82 (0.62–1.08)	HR of major hemorrhage-related deaths with aspirin vs. placebo: 1.13 (0.66–1.94)	HR of cancer-related deaths with aspirin vs. placebo: 1.31 (1.10–1.56)
Yusuf S. et al., 2021 [34]	Participants without cardiovascular disease who had an elevated INTERHEART Risk Score	6 years (mean 4.6 years)	5713	47.1/63.9	36.7%	HR of 3P-MACE ^a^ by aspirin vs. placebo: 0.86 (0.67–1.10)	No difference in major bleeding with aspirin vs. placebo (1.5% vs. 1.3%)	HR of cancer with aspirin vs. placebo: 0.83 (0.54–1.27)

Abbreviations: 3P-MACE, 3-point major adverse cardiovascular events; HR, hazard ratio; and RR, risk ratio. * Aspirin (100 mg) was compared to placebo or control therapy. ** Study drug was administered during the whole study period. *** Definition of cardiovascular events were different between the groups: ^a^ 3P-MACE: cardiovascular death, non-fatal, myocardial infarction, and stroke. ^b^ Composite of sudden death; death from coronary, cerebrovascular, and aortic causes; non-fatal acute myocardial infarction; unstable angina; newly developed exertional angina; nonfatal ischemic and hemorrhagic stroke; transient ischemic attack; or nonfatal aortic and peripheral vascular disease (arteriosclerosis obliterans, aortic dissection, mesenteric arterial thrombosis). ^c^ Death from coronary heart disease or stroke, non-fatal myocardial infarction or stroke, or above ankle amputation for critical limb ischemia. ^d^ Composite of nonfatal myocardial infarction, nonfatal stroke (excluding confirmed intracranial hemorrhage) or transient ischemic attack, or death from any vascular cause (excluding confirmed intracranial hemorrhage).

## Data Availability

The supporting data for the findings can be obtained from the corresponding author upon a reasonable request.

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
