# Peer review of "The Role of Cilostazol, a Phosphodiesterase-3 Inhibitor, in the Development of Atherosclerosis and Vascular Biology: A Review with Meta-Analysis"

_ijms, 2024, doi:10.3390/ijms25052593_

Round 1

Reviewer 1 Report

Comments and Suggestions for Authors

The article is very interesting, and presents a number of interesting novelties. However, there are points that need to be improved before it can be accepted:

1 - I didn't see the materials and methods of this article, being a systematic review article it should have the methods and put figure S2 in the body of the article and not as a supplement.

2 - the figure Risk-of-bias among trials included in the meta-analysis, S3 is very important, however it is not described how it was carried out, what programs were used, or the criteria?

3 - I don't understand Figure S1, where does the data come from? Is it from the authors? Or from an article? Are the authors obliged to publish this data?

Author Response

[Response to Reviewer #1’s comments]

The article is very interesting, and presents a number of interesting novelties. However, there are points that need to be improved before it can be accepted:

1 - I didn't see the materials and methods of this article, being a systematic review article it should have the methods and put figure S2 in the body of the article and not as a supplement.

→ Following the reviewer’s suggestion, we have comprehensively described the method part in the revised manuscript and moved Figure S2 to the main body.

[Revised] P. 4, lines 309-334 in the revised manuscript

4.1. Methods for meta-analysis of major adverse cardiovascular outcomes

For clinical interpretation, we systematically reviewed the effects of cilostazol on cardiovascular outcomes, as registered in PROSPERO CRD42023444959. We systematically searched PubMed, Embase, and the Cochrane databases for English-language studies published up to July 2023. The search terms included keywords such as 'cilostazol' for the study intervention; 'major adverse cardiovascular event', 'cardiovascular disease', 'cardiovascular event', 'myocardial infarction', and 'stroke' for the outcomes; and 'randomized controlled trial' for the study design. The full list of search terms can be found in Supplementary Table S1. Using this process, we selected the studies to be included in the present analysis based on the following inclusion criteria: studies that (1) compared cilostazol with placebo, control, aspirin, or clopidogrel; (2) reported composite major adverse cardiovascular events (MACEs); and (3) were RCTs. We excluded studies with a follow-up duration of less than 3 months.

The primary outcome was MACE. Secondary outcomes included individual cardiovascular events, such as fatal or non-fatal MI, fatal or non-fatal stroke, cardiovascular death, and all-cause mortality. Other adverse events, including major bleeding, discontinuation, headache, and palpitation, were also analyzed.

Study selection and data collection were confirmed by both authors. Logarithmic hazard ratios (HRs) were used to calculate combined estimates with a random-effects model. Random-effects risk ratios were calculated for studies that provided only the event numbers. The included trials were assessed for risk of bias using the Cochrane Risk of Bias 2 tool (see Supplementary Figure S3). The I2 statistic was employed to evaluate the overall heterogeneity of all comparisons. Publication bias was assessed using a funnel plot and Egger’s test. Sensitivity analyses were performed focusing on the regions of the trials. The meta-analysis was conducted using the 'metafor' statistical package in R software (version 4.1.0; R Development Core Team, Vienna, Austria).

2 - the figure Risk-of-bias among trials included in the meta-analysis, S3 is very important, however it is not described how it was carried out, what programs were used, or the criteria?

→ We have provided the methods for systematic review and the tool for analyzing publication bias.

[Revised] P. 4, lines 329-334 in the revised manuscript

The included trials were assessed for risk of bias using the Cochrane Risk of Bias 2 tool (see Supplementary Figure S3). The I2 statistic was employed to evaluate the overall heterogeneity of all comparisons. Publication bias was assessed using a funnel plot and Egger’s test. Sensitivity analyses were performed focusing on the regions of the trials. The meta-analysis was conducted using the 'metafor' statistical package in R software (version 4.1.0; R Development Core Team, Vienna, Austria).

3 - I don't understand Figure S1, where does the data come from? Is it from the authors? Or from an article? Are the authors obliged to publish this data?

→ The Figure S1 showed preliminary experimental study results conducted by our research team. However, this is not directly relevant to the content of the current manuscript. Therefore, we deleted it in the revised manuscript.

[Revised] P. 4, lines 294-295 in the revised manuscript

Cilostazol can increase uncoupling protein 1 (UCP1) in adipocytes [92].

Reviewer 2 Report

Comments and Suggestions for Authors

This is a comprehensive, well-written review on the subject.  I have very little to add.  Please consider the following comments.

1.     The title may be modified as we do not “combat” vascular biology.

2.     Lines 220-221.  “which is crucial for the proliferation of the neointima in blood vessels to a differentiated state.”  This is confusing to me.  Explain more or consider deleting.

3.     Lines 230-232.  “Cilostazol stimulates the production of hepatocyte growth factor, which is known for its role in regulating the growth, motility, and morphogenesis of endothelial cells.”  From the point of view of the healthy state of the endothelium, inducing endothelial cells to proliferate, migrate, and alter their phenotype does not yield positive outcomes.  In that sense, this statement is confusing.

4.     Lines 235-237.  “Moreover, cilostazol treatment improved the vasomotor reactivity response, as observed by an 80% change in the coronary luminal diameter after stimulation with N-monomethyl-l-arginine.”  This statement is difficult to comprehend.  More explanation is necessary.

5.     Line 296.  Throughout the text, please use the same unit for drug doses for the ease of comparison.  

Comments on the Quality of English Language

Some typos and punctuation errors.

Author Response

[Response to Reviewer #2’s comments]

This is a comprehensive, well-written review on the subject.  I have very little to add.  Please consider the following comments.

  1. The title may be modified as we do not “combat” vascular biology.

→ We thank the reviewer for this insightful comment. We have modified the current title.

[Revised] P. 1, lines 2-4 in the revised manuscript

The role of cilostazol, a phosphodiesterase-3 inhibitor, in the development of atherosclerosis and vascular biology: a review with meta-analysis

  1. Lines 220-221.  “which is crucial for the proliferation of the neointima in blood vessels to a differentiated state.”  This is confusing to me.  Explain more or consider deleting.

→ Following the reviewer's comment, we deleted this section in the revised manuscript to avoid potential confusion among readers.

  1. Lines 230-232.  “Cilostazol stimulates the production of hepatocyte growth factor, which is known for its role in regulating the growth, motility, and morphogenesis of endothelial cells.”  From the point of view of the healthy state of the endothelium, inducing endothelial cells to proliferate, migrate, and alter their phenotype does not yield positive outcomes.  In that sense, this statement is confusing.

→ We thank the reviewer for this insightful comment. In response, we have revised this part of the manuscript.

[Revised] P. 2, lines 229-231 in the revised manuscript

Cilostazol stimulates the production of hepatocyte growth factor, which is known for its favorable role in attenuating the malformation and apoptosis of endothelial cells [72].

  1. Lines 235-237.  “Moreover, cilostazol treatment improved the vasomotor reactivity response, as observed by an 80% change in the coronary luminal diameter after stimulation with N-monomethyl-l-arginine.”  This statement is difficult to comprehend.  More explanation is necessary.

 → As the reviewer suggested, we have revised this part to avoid any confusion in the manuscript.

[Revised] P. 2, lines 234-236 in the revised manuscript

Moreover, a 6-month treatment with cilostazol improved vasomotor reactivity, as evidenced by an increase in the coronary luminal diameter following administration of N-monomethyl-L-arginine [74].

  1. Line 296.  Throughout the text, please use the same unit for drug doses for the ease of comparison. 

→ We deleted the results of this experiment in the revised manuscript. Elsewhere, ‘mg’ units are used for cilostazol or aspirin throughout the manuscript.

[Revised] P. 6, lines 375-377 in the revised manuscript

A sustained-release form of cilostazol treatment (200 mg/day) for 12 months reduced the progression of coronary artery stenosis in patients with diabetes compared with aspirin (−3.1% vs. 1.7%).

Round 2

Reviewer 1 Report

Comments and Suggestions for Authors

This article was revised appropriately.

I recommend accept